# Functionalized Ordered Mesoporous MCM-48 Silica: Synthesis, Characterization and Adsorbent for CO_2_ Capture

**DOI:** 10.3390/ijms241210345

**Published:** 2023-06-19

**Authors:** Silvana Borcănescu, Alexandru Popa, Orsina Verdeș, Mariana Suba

**Affiliations:** “Coriolan Drăgulescu” Institute of Chemistry, Bl. Mihai Viteazul No. 24, 300223 Timisoara, Romania

**Keywords:** functionalized MCM-48 molecular sieve, amination reagents, temperature influence, CO_2_ adsorption, adsorption–desorption cycles

## Abstract

The ordered mesoporous silica MCM-48 with cubic Ia3d structure was synthesized using the cationic surfactant hexadecyltrimethylammonium bromide (CTAB) as a template agent and tetraethylorthosilicate (TEOS) as a silica source. The obtained material was first functionalized with (3-glycidyloxypropyl)trimethoxysilane (KH560); further, two types of amination reagents were used: ethylene diamine (N2) and diethylene triamine (N3). The modified amino-functionalized materials were characterized by powder X-ray diffraction (XRD) at low angles, infrared spectroscopy (FT-IR) and nitrogen adsorption–desorption experiments at 77 K. Characterization from a structural point of view reveals that the ordered MCM-48 mesoporous silica has a highly ordered structure and a large surface area (1466.059 m^2^/g) and pore volume (0.802 cm^3^/g). The amino-functionalized MCM-48 molecular sieves were tested for CO_2_ adsorption–desorption properties at different temperatures using thermal program desorption (TPD). Promising results for CO_2_ adsorption capacities were achieved for MCM-48 sil KH560-N3 at 30 °C. At 30 °C, the MCM-48 sil KH560-N3 sample has an adsorption capacity of 3.17 mmol CO_2_/g SiO_2_, and an efficiency of amino groups of 0.58 mmol CO_2_/mmolNH_2_. After nine adsorption–desorption cycles, the results suggest that the performance of the MCM-48 sil KH N2 and MCM-48 sil KH N3 adsorbents is relatively stable, presenting a low decrease in the adsorption capacity. The results reported in this paper for the investigated amino-functionalized molecular sieves as absorbents for CO_2_ can be considered as promising.

## 1. Introduction

Given the rapid industrialization increase and the sudden emission of CO_2_ into the atmosphere, it is necessary to constantly work on improving the adsorption properties of porous materials which are increasingly used for the adsorption of carbon dioxide. The main cause of CO_2_ emissions is the combustion of fuels (natural gas, coal and petroleum) for the production of transportation and electricity [1], which contributes to air pollution. Capture and sequestration of CO_2_ was considered as a way to reduce carbon dioxide emissions. Therefore, in recent years research on these porous materials has focused on the field of CO_2_ adsorption, as this method has proven to be more beneficial than conventional technologies. So far, a variety of solid-based materials such as zeolites, covalent organic frameworks, various functionalized mesoporous materials, etc., have been studied specifically for CO_2_ capture [2,3,4,5,6]. It is very important to develop effective methods for this purpose. Suitable adsorbents for CO_2_ should have several properties, such as good adsorption capacity, high selectivity and high stability [7].

MCM-48 mesoporous silica belongs to the M41s family and has a three-dimensional cubic Ia3d structure [8]. Based on previous X-ray diffraction analysis and transmission electron microscopy studies, MCM-48 mesoporous material was found to have a bicontinuous structure centered on the gyroid minimal surface [9], dividing the available pore space into two non-overlapping subvolumes [10]. Due to its structure and because possessing a large number of –OH groups is found to be an interesting material for CO_2_ adsorption and less liable to pore blocking, MCM-48 provides easy access to quest molecules provided from amines, carbonates or organosulfides. On the other hand, the synthesis of high-quality MCM-48 mesoporous silica is quite challenging, and is necessary to achieve specific adsorption and larger capacity [11,12]. MCM-48 materials, like the other M41s materials, have a large surface area (800–1400 m^2^ g^−1^), large pore volume and tunable pore size (2–50 nm). Due to its structural properties, mass transfer is faster compared to other mesoporous materials, which makes it a promising material for many other applications in the chemical field such as supports, adsorbents and catalysts [13,14,15,16,17].

Bandyopadhyay M. et. al. [13] synthesized MCM-48, MCM-41 and SBA-15 mesoporous silica and functionalized it with 3-aminopropyltriethoxysilane (APTES) for the transesterification reaction of triacetin with methanol. The obtained results showed that the catalytic activity of the studied mesoporous silica was significantly different. MCM-48 with NH_2_ showed an impressively good performance in the studied reaction (78% triacetin conversion) at 65 °C and a reaction time of 4.5 h. The good performance of the functionalized mesoporous MCM-48 material proved that it can be used as a potential catalyst for this reaction. Huang H.Y. and her team [18] concluded that MCM-48 and amine-surface-modified silica xerogel can be used for the selective adsorption of CO_2_ and H_2_S from natural gas streams. The excellent performances in terms of CO_2_ and H_2_S adsorption by the amine-modified MCM-48 are due to the large amounts of amine groups, the high surface areas and the high concentrations of silanol groups (SiOH) present on its surface. Mesoporous MCM-48 silica was prepared from rice husk ash (RHA), as a silica source, for CO_2_ capture applications at different temperatures (25, 50 and 75 °C), as reported by Jang H.T. et al. [19]. In this case, the amine-grafted sample, with 3-aminopropyltriethoxysilane (APTS) denoted APTS-MCM-48 (RHA), was compared with MCM-48 mesoporous silica prepared from RHA. APTS-MCM-48 (RHA) had a higher CO_2_ adsorption capacity (0.639 mmol CO_2_/g adsorbent) compared to MCM-48 (RHA) (0.033 mmol CO_2_/g adsorbent) at 25 °C adsorption temperature. These results suggest that MCM-48 mesoporous silica prepared from rice husk ash can be considered as a potential material for CO_2_ removal. Yismaw S. et al. [20] synthesized MCM-48 material by post-synthetic and co-condensation modification routes with 3-methacryloxypropyltrimethoxy-silane (MPS). The successful grafting was confirmed by the investigation method used and the cubic structure was fully preserved, as confirmed by the X-ray diffraction results. MCM-48 mesoporous material was synthesized and functionalized with aminopropyltriethoxy-silane (APTES) for carbon dioxide, as reported by Gil M. et al. [21]. After characterizing the designed material using different techniques, the results revealed a high efficiency for CO_2_ removal at low CO_2_ partial pressures, with loadings of 0.5 mmol CO_2_/molN at 5 kPa. The maximum adsorption capacity at 1 atm of CO_2_ was 1.68 mmol/g. Due to its interesting properties, Zhang Z. et al. [22] investigated double perovskites of Cs2AgBiBr6 (CABB) nanocrystals grown in situ in the MCM-48 mesoporous molecular sieve for photocatalytic CO_2_ reduction. The material is not only suitable for CO_2_ capture but also promotes CO_2_ activation by reducing the energy barriers.

In the present study, we focused on investigating how temperature influences the CO_2_ adsorption–desorption process on amino-functionalized MCM-48 materials. Different amine loadings and morphologies of MCM-48 functionalized with 3-glycidyloxypropyl trimethoxysilane (KH560) using two types of amination reagents, ethylene diamine (N2) and diethylene triamine (N3), were also investigated. To our knowledge, these amination reagents used to improve the MCM-48 adsorption–desorption properties as CO_2_ sorbent were not further mentioned in the literature for this type of material. Successful grafting on the MCM-48 mesoporous molecular sieve was confirmed by FT-IR, and the N_2_ adsorption–desorption investigations revealed a decrease in the pore volume and surface area of the functionalized mesoporous materials. The CO_2_ adsorption–desorption properties were investigated by temperature-programmed desorption combined with thermogravimetry: adsorption capacity (mg CO_2_/g adsorbent) and the efficiency of the amino groups used (mol CO_2_/mol NH_2_), respectively. These two characteristics are influenced by the type of amination agent and the amount of organic compound. The experimental results show very good values for both the adsorption capacity and amino group efficiency (ethylene diamine and diethylene triamine) of the studied MCM-48 mesoporous molecular materials. The gases that resulted during CO_2_ adsorption–desorption processes were identified by mass spectrometry coupled with the thermal analysis technique. The mesoporous functionalized materials aminated with ethylenediamine (N2) and diethylenetriamine (N3) were further tested in adsorption–desorption cycles to investigate their stability and regenerability in various practical applications for CO_2_ capture.

## 2. Results and Discussion

### 2.1. Physical—Chemical Characterization

The XRD patterns of the mesoporous molecular materials MCM-48 and MCM-48 sil KH are shown in Figure 1a. The XRD pattern of the investigated mesoporous molecular sieve MCM-48 (Figure 1) shows several peaks in the low angular range of 2Theta = 2–6 °Corresponding to the (211), (220), (420) and (332) reticular planes. The obtained XRD data of mesoporous molecular materials in correlation with literature data confirm the continuous cubic structure (Ia3d) of MCM-48 [8,10,23]. Functionalization of the MCM-48 mesoporous molecular sieve with KH560 does not affect the structural features of MCM-48, but its presence leads to a slight decrease in the peaks (Figure 1b). Figure 1b presents the XRD patterns of the samples aminated with ethylene diamine (N2) and diethylene triamine (N3). Increasing the number of amine molecules from two to three amine groups leads to a significant decrease in the intensity peaks characteristic for MCM-48 cubic structure. This behavior of the mesoporous molecular MCM-48 materials is because the characteristic peak intensities depend on the scattering contrast between the pore walls and pore channels, and usually decrease with a decrease in scattering contrast after the attachment of the organic group on the pore surface [13].

The FT-IR spectra of the MCM-48 molecular sieve and the amine-modified molecular sieve, MCM-48 sil KH, are shown in Figure 2a. The bare MCM-48 molecular sieve exhibited the same signals at 1082 cm^−1^, 805 cm^−1^ and 460 cm^−1^, which are characteristic of silica. The signals in the infrared spectra around 805 cm^−1^ and 460 cm^−1^ can be assigned to ν_s_ (Si–O–Si) and δ (Si–O–Si) bonds [19]. The signal around 1090 cm^−1^ attributed to ν_s_ (Si–O–Si) decreases somewhat due to the functionalization of MCM-48 with KH560 in the case of MCM-48 sil KH material. The same additional signals are observed in the FT-IR spectra: a broad, intense signal around 3440 cm^−1^ (ν O–H stretching) and a weaker absorption signal present around 1630 cm^−1^ (δH_2_O bending) indicating the presence of water. The signals around 2995 cm^−1^ correspond to the stretching of the C–H bonds of the CH_2_ and CH_3_ groups of KH560, and 1460 cm^−1^ can be attributed to the asymmetric deformation of the CH_3_-R bond. The FT-IR spectra of MCM-48 functionalized first with a silane coupling agent, KH560, and finally with two types of amination reagents, ethylene diamine (N2) and diethylene triamine (N3), are shown in Figure 2b. The difference between the MCM-48 sil KH material and the materials shown in Figure 2b is the presence of the vibrations characteristic for ethylene diamine and diethylene triamine. The signals around 1530–1465–1360 cm^−1^ on the FT-IR spectra [24,25] can be attributed to N–H stretching vibrations and N–H bending vibrations, respectively, and 956 cm^−1^ as asymmetric CH_3_–N^+^ stretching, being in accordance with the successful grafting of amines onto the MCM-48 mesoporous molecular sieve.

The specific surface area, pore volume and pore size distribution analyzed by BJH_Des_ method are presented in Table 1. The N_2_ adsorption–desorption isotherms of the MCM-48 mesoporous molecular sieve and of functionalized MCM-48 with KH560 and aminated with diethylene triamine are shown in Figure 3. The investigated materials present a typical reversible isotherm type IV according to IUPAC [25], which corresponds to mesoporous materials.

All the studied materials show an inflection at a relative pressure (p/p_0_) in the range of 0.1 to 0.4. This inflection reveals the stage of condensation of the capillary, which indicates a narrow range of uniform mesopores of these materials. As shown in Table 1, MCM-48, the mesoporous molecular sieve annealed at 550 °C, has the largest specific surface area (S_BET_ = 1466.5 m^2^/g) and pore volume (Vp = 0.802 cm^3^/g). In the case of the sample functionalized with the functionalizing agent KH560, the specific surface area and the pore volume decrease. This behavior of the functionalized material, MCM-48 sil KH, can be correlated with the filling of the pores with KH560, which is also confirmed by the FT-IR investigations. After grafting with N2 or N3 mesoporous molecular materials, the specific surface area decreases to 419 m^2^/g for MCM-48 sil KH N2 and to 505.3 m^2^/g for MCM-48 sil KH N3.

Scanning electron micrograph images were used to analyze the morphology of prepared mesoporous molecular sieves. In Appendix A, the morphology of the materials is studied by SEM images using different magnifications, more specifically 25,000× and 50,000×. The materials indicate that micrometric clusters are formed. It can be seen that by increasing the magnification from 25,000× up to 50,000×, the shape of the particles becomes more clear as spherical. The size of the spheres is around 200–500 nm.

It can be seen that the spherical morphology is still maintained after the introduction of KH560 and after N2 and N3 amine functionalization. After the functionalization of the materials, a change in the EDS spectra can be observed. So, from the EDS images presented in Appendix A, the existence of amino groups in the materials is confirmed by the presence of nitrogen peaks.

All four analyzed mesoporous molecular sieves, shown in Figure 4, show an initial endothermic effect on the DTA curves below 100 °C, with a mass loss of about 8÷12%, which is due to the release of physically sorbed water that overlaps with the water of crystallization. In the case of MCM-48, the decomposition of the functionalization agent used, KH560, occurs at about 260 °C (Figure 4b). The second mass loss, between 150 and 300 °C in the case of MCM-48 sil KH N2 (Figure 4c) and MCM-48 sil KH N3 (Figure 4d), must be due to the decomposition of the amination agent. The mass loss observed beyond 500 °C could be due to the dehydroxylation of the silica framework and/or the elimination of the residual ethoxy groups due to the incomplete hydrolysis of TEOS, as reported in the literature [26]. These results demonstrate the thermal stability of the investigated mesoporous MCM-48 materials. In agreement with these results, the TPD measurements were performed in a narrower temperature range, between 30 and 70 °C.

### 2.2. CO_2_ Adsorption–Desorption Measurements

The adsorption–desorption process of CO_2_ by thermogravimetry was carried out for amino-functionalized molecular sieves using the TPD program. Figure 5 shows the steps of the pre-treated samples with adsorption–desorption processes for MCM-48 sil KH N2 (Figure 5a) and MCM-48 sil KH N3 (Figure 5b) in the temperature test range 30–70 °C.

The developmental steps of the adsorption–desorption process are presented as follows: in the first step, each investigated sample was pre-treated in flowing N_2_ at 150 °C and kept at this temperature for half an hour. During this first step, a constant sample mass was achieved as the N_2_ stream flowed over the surface of the amino-functionalized molecular sieve under study, cleaning it of all organic contaminants and adsorbed gases from the environment.

In the second step, the temperature was lowered to the studied adsorption temperature of 30, 40, 50 and 70 °C and kept under N_2_ atmosphere for another additional 30 min. The sample was then exposed to the adsorption gas mixture of 30% CO_2_/N_2_ (70 mL/min) and held for 90 min. After this step, the sample was kept in a N_2_ atmosphere at the same temperature for another 30 min to remove most of the physisorbed CO_2_. The desorption step of the chemisorbed CO_2_ from the amino-functionalized molecular sieve was carried out in the range from the adsorption temperature to 180 °C, with an increasing temperature rate of 10 °C/min and with an isotherm at 180 °C for 30 min. All the steps required to perform a single CO_2_ adsorption/desorption process take about 4 h.

Figure 5c,d presents the evolution of CO_2_ by MS spectra during the adsorption–desorption process. During the gas CO_2_/N_2_ mixture exposure, the signal of CO_2_ increases and it is maintained relatively constant for 90 min. Stopping the gas mixture for 30 min, the signal of CO_2_ decreases while the samples are maintained in the N_2_ atmosphere. The CO_2_ signal increases again—selected area in the graphs of Figure 5c,d—during the desorption step. CO_2_ is the main compound present on the MS spectra.

Two main parameters of the adsorption–desorption process were calculated from the adsorption isotherms shown in Figure 5: the adsorption capacity of the adsorbent, measured in mmol CO_2_ per gram of adsorbent, and the adsorption efficiency of the adsorbent, measured in mmol CO_2_/mmol NH_2_. The formulae used to calculate the above parameters are published elsewhere [27,28]. Figure 6 shows the amounts of CO_2_ captured for MCM-48 sil KH N2 (Figure 6a) and MCM-48 sil KH N3 (Figure 6b). The influence of temperature on the investigated mesoporous molecular sieves is practically the same; the amounts of CO_2_ absorbed decrease with increasing the temperature from 30 °C to 70 °C. The optimal temperature for both MCM-48 sil KH N2 and MCM-48 sil KH N3 seems to be 30÷50 °C according to the obtained results. In the case of MCM-48 sil KH N2, very good results were obtained, both for the adsorption capacity (2.92 mmol CO_2_/g SiO_2_) and the amino group efficiency (0.85 mmol CO_2_/mmol NH_2_). Higher values for adsorption capacity (3.17 mmol CO_2_/g SiO_2_) and amino group efficiency (1.17 mmol CO_2_/mmol NH_2_) were obtained for the mesoporous molecular sieve MCM-48 sil KH N3. These values decrease at 50 °C as follows: 2.64 mmol CO_2_/g SiO_2_ and 0.76 mmol CO_2_/mmol NH_2_ for MCM-48 sil KH N2, and 2.79 mmol CO_2_/g SiO_2_ and 0.99 mmol CO_2_/mmol NH_2_, respectively, (Figure 6).

The optimal temperature for CO_2_ adsorption depends on the amine and support type used. The reaction mechanism between primary or secondary amino groups and CO_2_ in an anhydrous environment, which acts like a base, leads to the formation of carbamates, as the final product, due to the formation of zwitterions between the amino group and the carbon dioxide molecule [29]. The chemical interaction between CO_2_ and amine can be expressed as follows:R-NH_2_ + CO_2_ → R-NH_2_^+^COO^−^(1)
R-NH_2_^+^COO^−^ + R-NH_2_ → R-NHCOO^−^ + RNH_3_^+^(2)

While desorption of CO_2_ and regeneration of grafted mesoporous silica is expressed as follows:R-NH-COO^−^ + R-NH_3_^+^ + Heat → CO_2_ + 2R-NH_2_(3)

The formation of the carbamate involves equilibrium; therefore, it is reversible. This reaction mechanism proposed by Caplow [30] corresponds to chemisorption, but at higher partial pressure physisorption also takes place [31]. Increasing the temperature to 70 °C leads to smaller values for both investigated parameters for MCM-48 sil KH N2 (adsorption capacity 1.76 mmol CO_2_/g SiO_2_ and amino group efficiency 0.50 mmol CO_2_/mmol NH_2_) and MCM-48 sil KH N3 (adsorption capacity 1.51 mmol CO_2_/g SiO_2_ and amino group efficiency 0.56 mmol CO_2_/mmol NH_2_).

Figure 7 shows the evolution of CO_2_ uptake with time (Figure 7a,c) and its time derivative (Figure 7b,d) for the adsorbents MCM-48 sil KH N2 and MCM-48 sil KH N3. The CO_2_ adsorption rate for the investigated adsorbents MCM-48 sil KH N2 and MCM-48 sil KH N3 is in the range of 0.4 ÷ 1.4 mmol CO_2_/g SiO_2_·min depending on the temperature treatment at 30 ÷ 70 °C.

Helen Y. Huang and her coworkers studied the adsorption of CO_2_ and H_2_S on the MCM-48 and silica xerogels amine-modified surface using 3-aminopropyltrietoxy-silane [18] by the method developed by Schumacher [32]. They conclude that the adsorbed amount of CO_2_ increases with the increase in partial pressure from 0 to 0.1 atm. CO_2_ adsorbed on the amine-modified MCM-48 is equivalent to 2.05 mmol/g at 1 atm. Bhagiyalakshmi M. et al. investigated the use of 3-chloropropyl amine hydrochloride (3-CPA) as a grafting agent on MCM-48 mesoporous material synthesized from RHA, as the silica source. The sample noted MCM-48/CPA presents a 1.1 mmol/g CO_2_ adsorption capacity [33]. According to the published results by Bhagiyalakshmi, 3-CPA can be used as an adsorbent for CO_2_. Qian X. et al. reported that PEI-based pore expended adsorbent, 40% loading on MCM-48-W mesoporous molecular material, exhibited 2.59 mmol/g of CO_2_ adsorption capacity (6.9 mmol CO_2_/g PEI) with excellent regeneration ability [34].

Some of the results reported in the literature regarding the adsorption–desorption performance of the mesoporous molecular materials with different amine loadings are presented in Table 2.

One of the most important parameters in the evaluation of the investigated mesoporous molecular materials as sorbents for CO_2_ capture is the multi-cycle stability tests for a longer operating time. For this purpose, nine cycles of CO_2_ adsorption–desorption thermal measurements were performed for MCM-48 sil KH N2 and MCM-48 sil KH N3-grafted mesoporous molecular sieves with a relatively good adsorption amount of CO_2_ for standing samples, 2.92 mmol CO_2_/g SiO_2_ and 3.17 mmol CO_2_/g SiO_2_, respectively. The samples were pre-treated in flowing nitrogen at 120 °C for 10 min, then cooled to the adsorption temperature of 30 °C and exposed to 30% CO_2_ in N_2_ for 40 min. For the desorption test, the samples were heated to 120 °C with 10 °C/min. The samples tested over repeated adsorption–desorption stability tests are shown in Figure 8.

The adsorption capacity decreases after the nine adsorption–desorption cycles, from 4.44 to 4.08 mmol CO_2_/g SiO_2_ for MCM-48 sil KH N2 and from 5.41 to 4.75 mmol CO_2_/g SiO_2_ for MCM-48 sil KH N3, with a better CO_2_ adsorption capacity observed for the sample grafted with N3. Upon closer examination of the samples via repeated adsorption–desorption stability tests, both MCM-48 sil KH N2 and MCM-48 sil KH N3 show a slight decrease in adsorption capacity. The regeneration stability tests showed a relatively good performance of the investigated mesoporous molecular material for a possible industrial application.

## 3. Methods and Materials

### 3.1. Preparation of the Samples

The synthesis of MCM-48 was prepared according to the method developed by Ortiz [37] and his research team. As part of a brief description of the preparation of MCM-48 mesoporous material, the following chemical materials were used: cetyltrimethylammonium bromide CTAB (98%, Fluka, Biochemika, Charlotte, NC, USA) was used as the structure directing agent; tetraethylorthosilicate (TEOS, ≥99% Merck, Darmstadt, Germany) was used as a source of silica; deionized water, ethanol (C_2_H_5_OH, 99.2% Chimreactiv, București, Romania) and aqueous ammonia solution (NH_4_OH, 20% Chimreactiv, București, Romania) were used as reagents for the synthesis. In addition, (3-glycidyloxypropyl)trimethoxysilane for the amino-functionalized molecular sieves (KH560 Sigma-Aldrich, Steinheim, Saint Louis, MO, USA) was used as a silane coupling agent and two types of amination agents were used: ethylene diamine (N2) (C_2_H_8_N_2_, Merck, Darmstadt, Germany) and diethylene triamine (N3) (C_4_H_13_N_3_, Merck, Darmstadt, Germany). In a typical synthesis of the mesoporous MCM-48 molecular sieve, 5.2 g CTAB was added to 240 mL deionized water and 100 mL ethanol under continuous stirring. After the CTAB had dissolved and the solution had become clear, 24 mL of NH_4_OH was added to the system and mixed for a few minutes. Then, 7.2 mL of TEOS was immediately poured into the solution under vigorous stirring. Stirring was continued for 15 h at room temperature. The solid white product was recovered by filtration and dried at room temperature overnight. The dried materials were annealed at 540 °C for 6 h to remove the CTAB from the composite material.

For the functionalization of the MCM-48 molecular sieve with KH560, denoted MCM-48 sil KH, 1 g of MCM-48 was added to a solution containing 100 mL of toluene and after stirring for a few minutes, 4 mL of (3-glycidyloxypropyl)trimethoxysilane was added. The resulting mixture was refluxed at 110 °C for 6 h. The mixture was then collected by filtration, washed with ethanol and dried in an oven at 80 °C for 4 h. For the amination process, 3 mL of amination reagent was added to the dispersed system of MCM-48 sil KH and 100 mL of toluene. The grafting reaction was carried out at 110 °C for 6 h. After filtration and drying, the absorbents were obtained as white solids. The samples were named MCM-48 sil KH N2 and MCM-48 sil KH N3.

### 3.2. Characterization of the Samples

The prepared molecular sieve MCM-48 and the functionalized samples were characterized by the following investigation methods: FT-IR spectroscopy, X-ray diffraction analysis, BET specific surface area and thermal analysis.

The FT-IR absorption spectra were recorded with a Jasco 430 spectrometer (Tokyo, Japan) in the range 4000–400 cm^−1^, using KBr pellets. X-ray diffraction analyses were performed with Bruker D8 Advance X-ray diffractometer (Karlsruhe, Germany), equipped with a copper target X-ray tube in the range of small angle 2 Theta = 0.5–5.

Textural characteristics of the outgassed samples were determined from nitrogen physisorption at 77 K using a Quantachrome instrument: the Nova 2000 series instrument (Boynton Beach, FL, USA). The specific surface area S_BET_, the average pore diameter BJH_Des_ and the adsorption pore volume VpN_2_ were determined.

The resulting functionalized materials were characterized by thermal analysis using a TGA/SDTA 851-LF 1100 instrument from Mettler (Columbus, OH, USA). The thermal analysis system was coupled to a Pfeiffer—Vacuum—Thermo Star mass spectrometer via a silica capillary at a temperature of 200 °C. Adsorption measurements were performed using the same thermogravimetric analyser connected to a gas feed manifold. High-purity CO_2_ and 30% CO_2_ in N_2_ at 1 atm were used for the adsorption runs, and N_2_ was used as the regenerating purge gas for CO_2_ desorption. In a typical adsorption–desorption run, a blank test of the empty sample container was performed at 25 °C in a N_2_ stream at a flow rate of 50 mL min^−1^. The samples were then weighed and placed in the sample container. An amount of 20 mg of each sample was placed in an aluminum crucible of 150 µL. Measurements were carried out in a dynamic air atmosphere with a flow rate of 50 mL min^−1^, in the temperature range of 25–650 °C with a heating rate of 10 °C min^−1^. Each sample was first pre-treated in flowing N_2_ at 150 °C, then cooled to the desired adsorption temperature (30–70 °C), and exposed to a 30% gas mixture CO_2_/N_2_ (70 mL min^−1^) for 120 min. The CO_2_ adsorption capacity of the adsorbent in milligrams of CO_2_ per gram of adsorbent was calculated from the mass gain of the sample during the adsorption process. The most promising results for the tested adsorbents were investigated based on the stability behavior and the ability to regenerate through adsorption–desorption cycles for efficient regeneration.

## 4. Conclusions

The MCM-48 mesoporous molecular sieve with cubic Ia3d structure functionalized with (3-glycidyloxypropyl)trimethoxysilane (KH560) and various amine loadings was studied. The formation of the cubic structure of MCM-48 with a large specific surface area of 1466.5 m^2^/g is confirmed by combined XRD crystal structure test and Fourier infrared spectra analysis.

The functionalization of KH560 on the MCM-48 molecular sieve is evident from the presence of the characteristic bands on the FT-IR spectra. The successful grafting of ethylene diamine and diethylene triamine is confirmed by FT-IR analysis and by N_2_ adsorption–desorption isotherms, which showed a significant decrease in specific surface area to 419 m^2^/g for MCM-48 sil KH N2 and 505.3 m^2^/g for MCM-48 sil KH N3.

By loading the amine on the MCM-48 molecular sieve, the –NH_2_ functional group of N2 and N3 acts as an active site, leading to the formation of a porous microstructure that facilitates the diffusion of CO_2_ into the material structure and its adsorption.

The CO_2_ adsorption–desorption tests on the studied samples at temperatures between 30 and 70 °C by thermogravimetric method and temperature-programmed desorption showed that CO_2_ adsorption strongly depends on textural properties. Good results in terms of CO_2_ adsorption were obtained for MCM-48 sil KH N2 and MCM-48 sil KH N3 studied at 30, 40 and 50 °C. The best results were obtained at 30 °C for the MCM-48 sil KH N3 mesoporous molecular sieve: 3.17 mmol CO_2_/g SiO_2_ for adsorption capacity and 1.17 mmol CO_2_/mmol NH_2_ for amino group efficiency, which are better than other results reported in the literature.

The slight decrease in regeneration stability tests showed a relatively good performance of the investigated mesoporous molecular adsorbents, making them a promising material for future industrial CO_2_ capture applications.

## Figures and Tables

**Figure 1 ijms-24-10345-f001:**
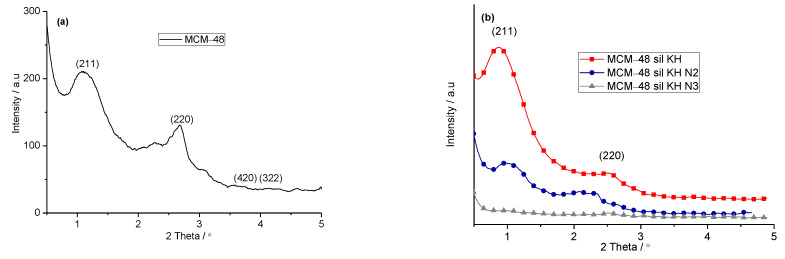
XRD patterns of MCM-48 (**a**) and MCM-48 sil KH, MCM-48 sil KH N2 and MCM-48 sil KH N3 mesoporous molecular materials (**b**).

**Figure 2 ijms-24-10345-f002:**
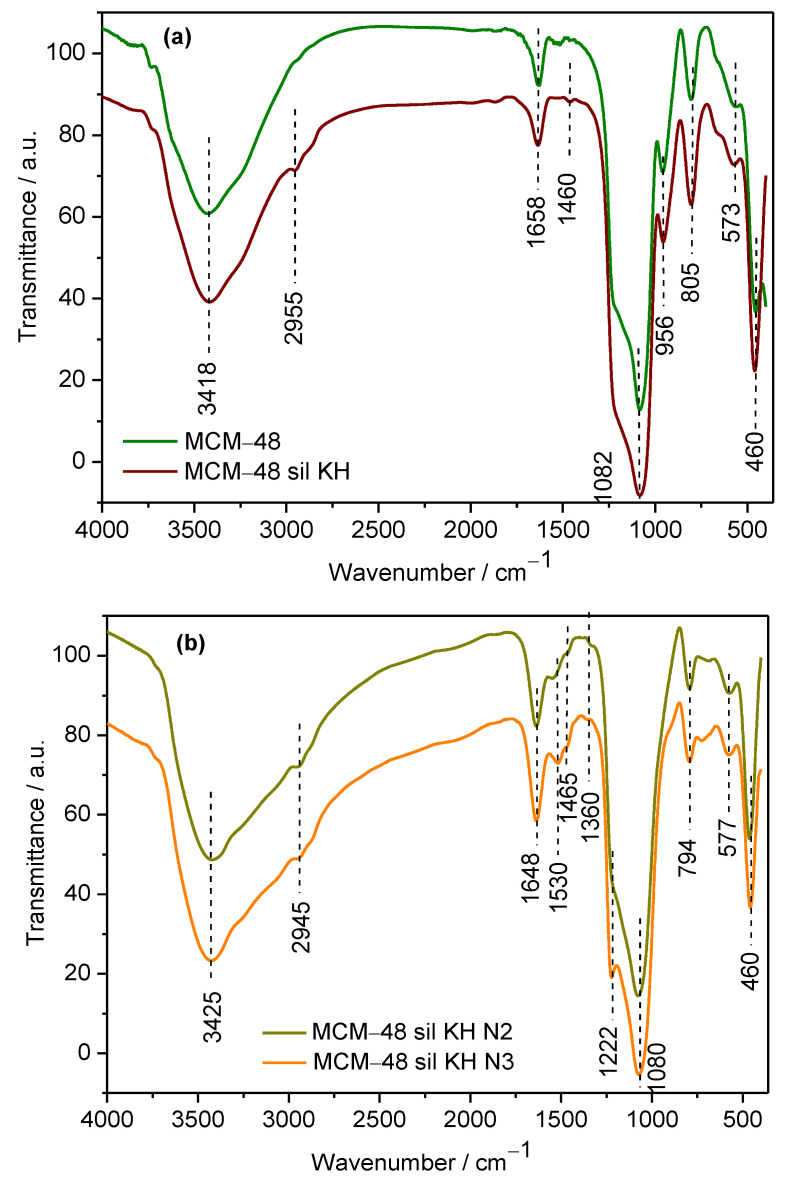
FT-IR spectra of MCM-48 and MCM-48 sil KH (**a**) and MCM-48 sil KH N2 and MCM-48 sil KH N3 (**b**) mesoporous materials.

**Figure 3 ijms-24-10345-f003:**
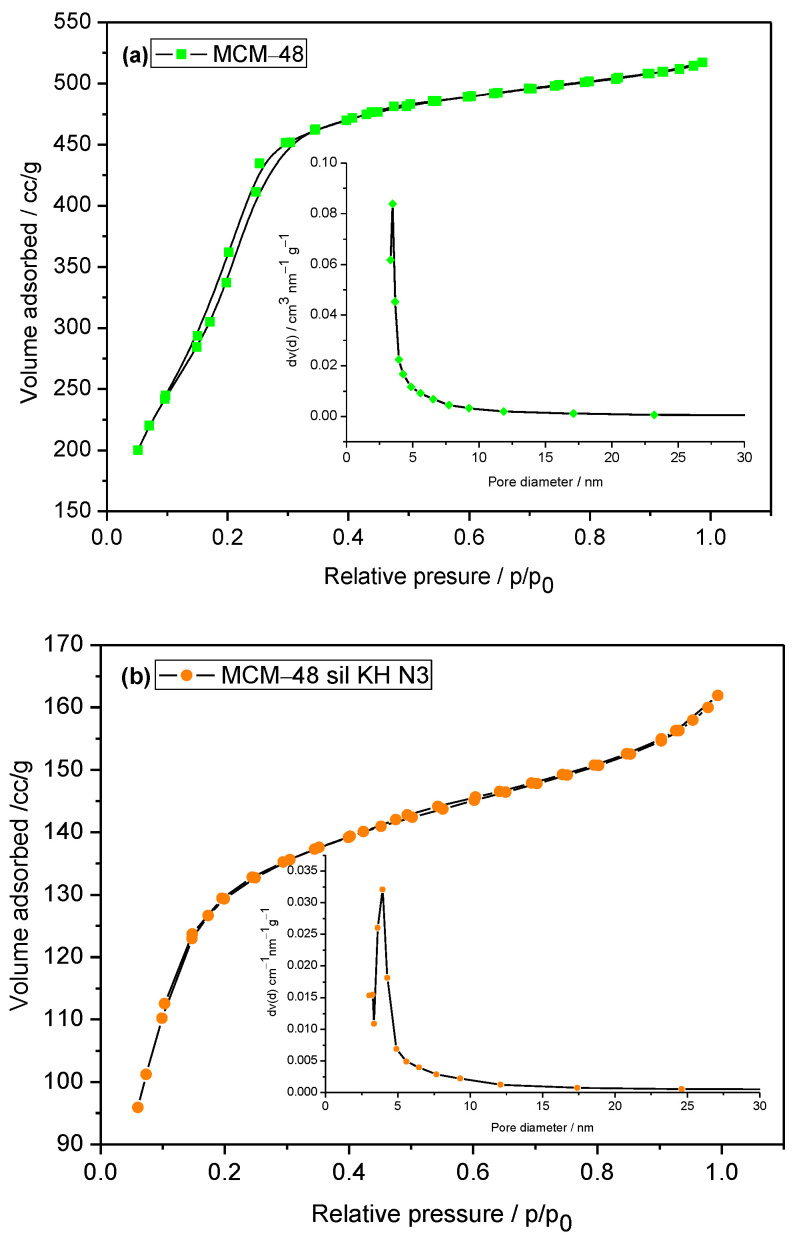
Nitrogen adsorption–desorption isotherms and the pore size distribution of MCM-48 (**a**) and MCM-48 sil KH N3 (**b**).

**Figure 4 ijms-24-10345-f004:**
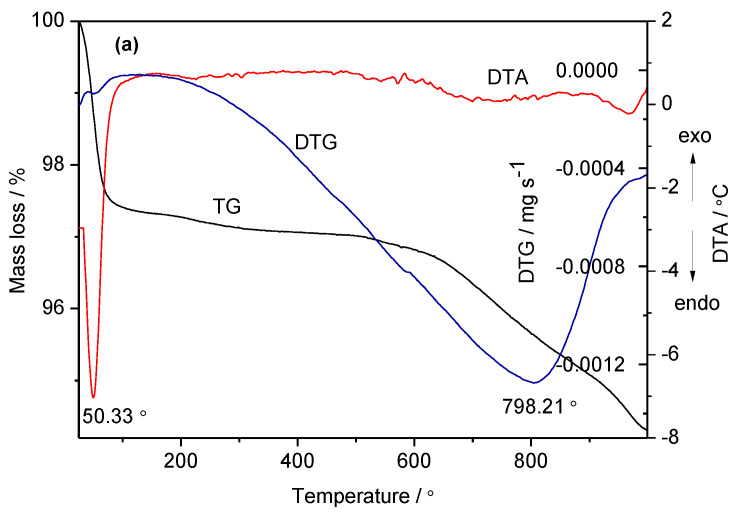
TG-DTA-DTG curves of MCM-48 (**a**), MCM-48 sil KH (**b**), MCM-48 sil KH N2 (**c**) and MCM-48 sil KH N3 (**d**) mesoporous materials.

**Figure 5 ijms-24-10345-f005:**
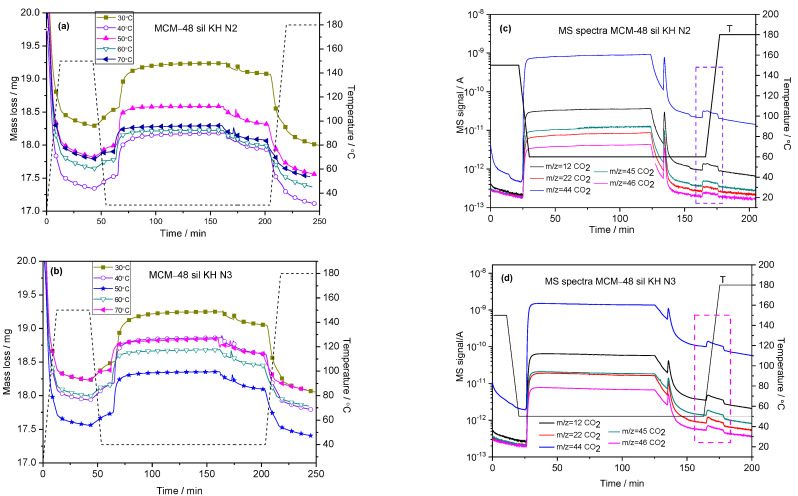
CO_2_ adsorption–desorption steps of functionalized samples MCM-48 sil KH N2 (**a**) and MCM-48 sil KH N3 (**b**) with an adsorption isotherm at 30–70 °C and MS spectra for the MCM-48 sil KH N2 sample at 40 °C (**c**) and the MCM-48 sil KH N3 sample at 40 °C (**d**).

**Figure 6 ijms-24-10345-f006:**
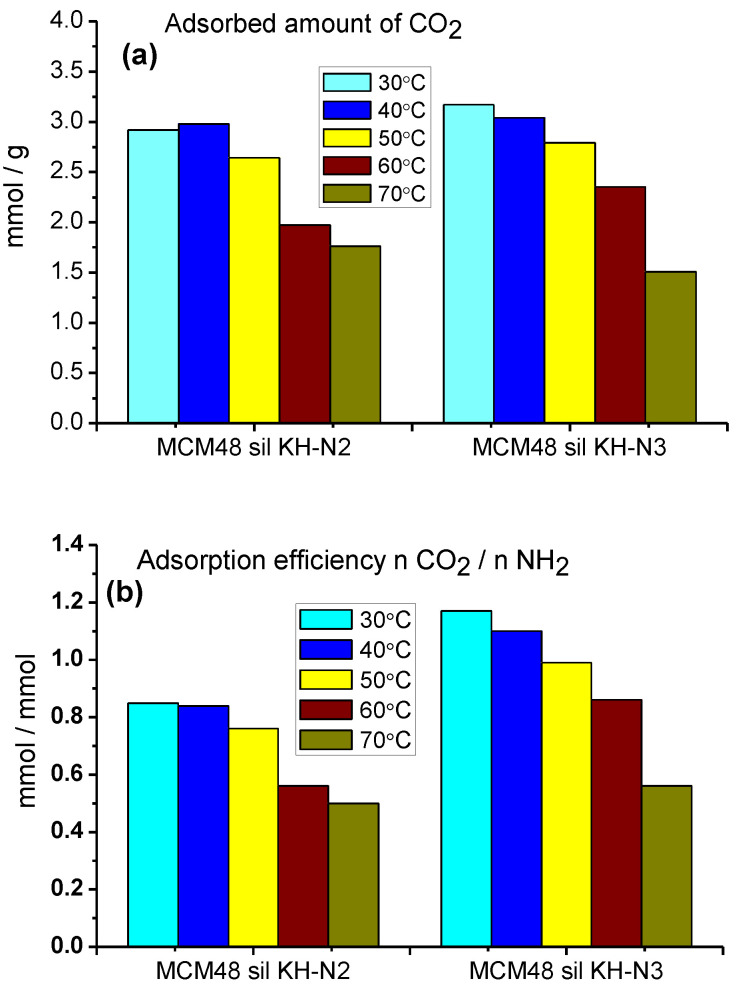
The absorbed amounts of the captured CO_2_ on MCM-48 sil KH N2 and MCM-48 sil KH N3 (**a**) and their adsorption efficiency (**b**) at different temperatures.

**Figure 7 ijms-24-10345-f007:**
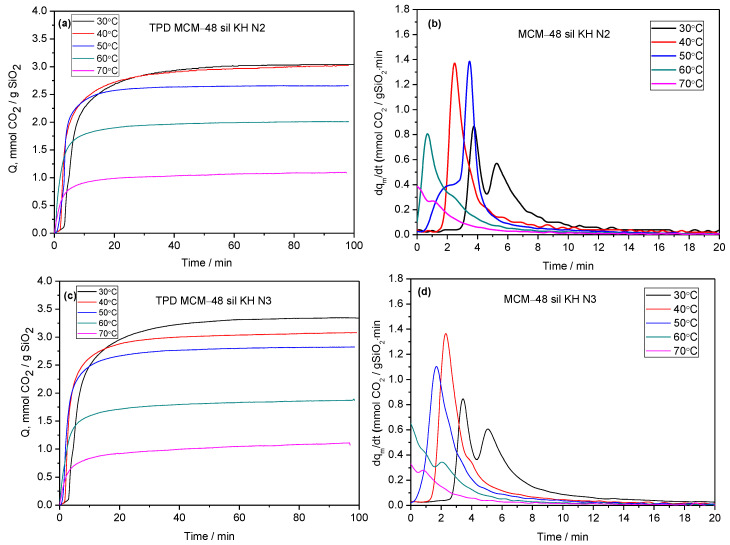
Carbon dioxide adsorption (**a**,**c**) and carbon dioxide adsorption rate (**b**,**d**) of MCM-48 sil KH N2 and MCM-48 sil KH N3 at temperatures between 30 and 70 °C.

**Figure 8 ijms-24-10345-f008:**
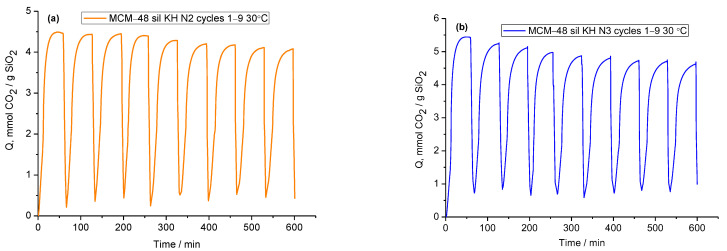
CO_2_ adsorption–desorption cycles measurements for MCM-48 sil KH N2 (**a**) and MCM-48 sil KH N3 (**b**) with adsorption at 30 °C.

**Table 1 ijms-24-10345-t001:** Textural properties of the MCM-48 molecular sieve and its derivate mesoporous materials.

No.	Sample	Specific Surface Area (m^2^/g)	Pore Volume BJH_Des_ (cc/g)	Average Pore Diameter BJH_Des_ (nm)
1.	MCM-48	1466.5	0.802	3.48
2.	MCM-48 sil KH	652	0.295	3.91
3.	MCM-48 sil KH N2	419	0.211	3.93
4.	MCM-48 sil KH N3	505.3	0.251	3.23

**Table 2 ijms-24-10345-t002:** Adsorption–desorption performance of the mesoporous materials with different amine type loadings.

Support	Amine Type	Adsorption Conditions	CO_2_ Adsorption Capacity (mmol CO_2_/g ads)	Reference
MCM-48	APTES	25 °C, up to 1 bar CO_2_	2.05	Huang et al. [18]
SBA-15	CPA	25 °C, up to 1 bar CO_2_	1.50	Bhagiyalakshmi et al. [32]
MCM-48			1.10	
MCM-41			1.70	
Si-MCM-41	MEA DEA TEA	25 °C, up to 1 bar pure CO_2_	0.46–0.65	Mello et al. [35]
MCM-41	APTES	20 °C, up to 1 bar CO_2_	0.78	Mukherjee et al. [36]
	MEA	25–60 °C, up to 1 bar CO_2_	1.71	
	BZA		0.66	
	AEEA		1.07	
MCM-48	PEI	25 °C, up to 1 bar pure CO_2_	2.59	Qian et al. (2019) [34]
MCM-48 sil KH N2	EDA	30 °C, up to 1 bar CO_2_	2.92	This work
MCM-48 sil KH N3	DETA	30 °C, up to 1 bar CO_2_	3.17	This work

## Data Availability

Not applicable.

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
