# Peer review of "Functionalized Ordered Mesoporous MCM-48 Silica: Synthesis, Characterization and Adsorbent for CO2 Capture"

_ijms, 2023, doi:10.3390/ijms241210345_

Round 1

Reviewer 1 Report (Previous Reviewer 1)

The paper can be accepted.

Author Response

Reviewer 2 Report (Previous Reviewer 2)

The anthors have completely improved the equality of their paper.  this manuscript  can be published without further modification.

Author Response

Reviewer 3 Report (New Reviewer)

Popa et al. focused their attention on the investigation to find out how the temperature influences the CO2 adsorption-desorption process on amino functionalized MCM-48 materials. The topic is very interesting but there are some issues that can be solved:

1. English polishing is required;

2. Correct the typos;

3.  Correct the underlined sentences;

4. The results are repeatable? Please report the standard deviations.

5. The difference between the CO2 adsorption capacity of this article and the one obtained by Qian et al. seem to be very low. Is this difference significative? Please explain well the advantage of your achievements in your conclusion.

6. SEM analysis are missing. 

1. English polishing is required;

2. Correct the typos;

3.  Correct the underlined sentences.

Author Response

Reviewer 4 Report (New Reviewer)

The article is devoted to the synthesis of mesoporous silica  functionalized with amine-containing molecules to enhance the sorption properties for CO2. The synthesized samples were characterized by BET, FTIR, XRD, and the CO2 sorption properties were tested by the gravimetric method with the support of mass spectrometry. The article leaves a good impression, the discussions are quite detailed. There are minor remarks:

1. Part of the text for some reason is written in underlined italics; a uniform plain text is required for publication.

2. Equations 1-2 are written incorrectly. The rules for writing chemical equations require that there be a mass and charge balance of the left and right sides. However, in equation (1), the left side includes neutral particles, and the right side includes one negatively charged one (COO-). Similarly, the note of charge imbalance applies to equation (2). Please fix this or provide an explanation.

English is understandable enough.

Round 2

Reviewer 3 Report (New Reviewer)

Dear authors, the responses to my comments are fine.

This manuscript is a resubmission of an earlier submission. The following is a list of the peer review reports and author responses from that submission.

Round 1

Reviewer 1 Report

Alexandru Popa et al. synthesized the ordered mesoporous MCM-48 and used for CO2 capture. The manuscript reported some interesting results and is within the scope of publication of the journal. However, the paper should undergo minor revisions before publication:

1. Introduction section does not show the comprehensive research progress of the glycerol applications and tungstophosphoric acid catalysts. The background knowledge on this area needs to be further reviewed, by consulting the recently published articles, for example: Journal of Industrial and Engineering Chemistry, 2023, 117, 85-102; Chemical Engineering Journal Volume 316, 15 May 2017, Pages 797-806.

2. The authors should list a table to compare their results with published literatures.

3. The reaction mechanism should be added.

Author Response

Please see then attachment

Reviewer 2 Report

The present version resubmitted by the authors has been greatly improved compared to the previous manuscript. Authors have addressed my concerns. Therefore, I recommend for pubulication of this paper without further modification 

Reviewer 3 Report

The manuscript develops a series of amine-functionalized ordered mesoporous MCM-48 silicas for CO2 capture. Although these amine-functionalized MCM-48 displays a moderate CO2 adsorption capacity, there are serious problems with the writing and drawing of this article. Hence, this article is not suitable for publication in Int. J. Mol. Sci.

1. Relevant XRD data of MCM-48 sil KH N2 and MCM-48 sil N3 are missing?

2. What is the meaning of “2θ=2°÷6°” in Line 102? Line 145... 

3. The Figure number in line 107 is wrong. 

4. All the drawings need to be redone.

Reviewer 4 Report

The manuscript reads well and the topic is of importance.

I am a bit surprised to read the valus of the Specific Surface Area (m2/g) and the Pore Volume BJHDes (cc/g) in the abstract and in Table 1. While the specific surface area (in the reange of 400 to 1400) compares well with recent reports regarding  MCM-41 and SBA-15 values (ref [1]: Microporous Mesoporous Materials, 312, 2021, 110744, 1-12. https://doi.org/10.1016/j.micromeso.2020.110744 and ref [2]: Material Chemistry and Physics 296, 2023, 127121, 1-8. https://doi.org/10.1016/j.matchemphys.2022.127121) the values of the pore volume (in the range of 0.2 to 0.8) is 50 to 100 time smaller than what I would have expected.

This is strange. Also, because the reported values for the pore diameter are similar to those found in refs [1 and 2].

I strongly recommend that the authors discuss their Surface Area and the Pore Volume BJHDes data with respect to the values reported in refs [1 and 2].

I would also appreciate a comparison of the synthesis procedure with the procedure reported in ref [3]: D. Brühwiler, H. Frei, J. Phys. Chem. B 107 (2003) 8547–8556.

Otherwise the manuscript is ok, in my opinion.

Round 2

Reviewer 3 Report

The paper does not meet the requirements for journal publication. The Figures ares the same as last time, no changes have been made. As a result, the paper is not suitable to publish in International Journal of Molecular Science.

Reviewer 4 Report

The authors have not sufficiently improved the manuscript.

This is not a careful revision and demonstrates low quality work.

Their response to the reviewer comments is insufficient in some aspects not correct.

The manuscript is not ready for publication.

I recommend rejection and resubmission of a completely revised manuscript which should be considered as a new submission.